# Ark Shell-Derived Peptides AWLNH (P3) and PHDL (P4) Mitigate Foam Cell Formation by Modulating Cholesterol Metabolism and HO-1/Nrf2-Mediated Oxidative Stress in Atherosclerosis

**DOI:** 10.3390/md23030111

**Published:** 2025-03-05

**Authors:** Chathuri Kaushalya Marasinghe, Jae-Young Je

**Affiliations:** 1Department of Food and Life Sciences, Pukyong National University, Busan 48513, Republic of Korea; chathurimarasinghe9313@gmail.com; 2Major of Human Bioconvergence, Division of Smart Healthcare, Pukyong National University, Busan 48513, Republic of Korea

**Keywords:** blue mussel, bioactive peptides, RAW264.7 macrophages, atherosclerosis, inflammation, cholesterol flux, oxidative stress

## Abstract

Atherosclerosis, a leading contributor to cardiovascular diseases (CVDs), is characterized by foam cell formation driven by excessive lipid accumulation in macrophages and vascular smooth muscle cells. This study elucidates the anti-atherosclerotic potential of AWLNH (P3) and PHDL (P4) peptides by assessing their effects on foam cell formation, lipid metabolism, and oxidative stress regulation. P3 and P4 effectively suppressed intracellular lipid accumulation in RAW264.7 macrophages and human aortic smooth muscle cells (hASMCs), thereby mitigating foam cell formation. Mechanistically, both peptides modulated cholesterol homeostasis by downregulating cholesterol influx mediators, cluster of differentiation 36 (CD36), and class A1 scavenger receptor (SR-A1), while upregulating cholesterol efflux transporters ATP-binding cassette subfamily A member 1 (ABCA1) and ATP-binding cassette subfamily G member 1 (ABCG1). The activation of peroxisome proliferator-activated receptor-gamma (PPAR-γ) and liver X receptor-alpha (LXR-α) further substantiated their role in promoting cholesterol efflux and restoring lipid homeostasis. Additionally, P3 and P4 peptides exhibited potent antioxidative properties by attenuating reactive oxygen species (ROS) generation through activation of the HO-1/Nrf2 signaling axis. HO-1 silencing via siRNA transfection abolished these effects, confirming HO-1-dependent regulation of oxidative stress and lipid metabolism. Collectively, these findings highlight P3 and P4 peptides as promising therapeutic agents for atherosclerosis by concurrently targeting foam cell formation, cholesterol dysregulation, and oxidative stress, warranting further exploration for potential clinical applications.

## 1. Introduction

Cardiovascular diseases (CVDs), a leading cause of mortality worldwide, encompass a broad spectrum of conditions, including coronary artery disease, cerebrovascular disease, and deep vein thrombosis [1,2]. These conditions are primarily characterized by the gradual accumulation of lipids within arterial walls, leading to the narrowing or complete blockage of blood flow to critical organs such as the heart and brain, ultimately resulting in myocardial infarction (MI) or stroke [3]. The underlying pathology of most CVDs is atherosclerosis, with foam cell formation being a pivotal process in its development [4,5]. Foam cells arise when monocytes migrate into the subendothelial space, differentiate into macrophages, and engulf modified lipoproteins, particularly oxidized low-density lipoproteins (oxLDLs) [6]. This process disrupts cholesterol homeostasis, as oxLDLs promote excessive cholesterol influx and hamper cholesterol efflux, triggering the progression of atherosclerosis and subsequent CVDs [4]. Since foam cell formation is central to every stage of atherosclerosis, from plaque initiation and growth to eventual rupture [4], consequently, a deeper understanding of the mechanisms driving foam cell formation has become a focal point for developing strategies to combat atherosclerosis and mitigate CVD risk. This area continues to attract significant scientific interest due to its critical implications for global health.

In recent decades, bioactive natural substances have emerged as promising therapeutic agents, owing to their diverse composition of biologically active components [7]. Aquatic environments, which cover approximately 71% of the Earth’s surface, host unique biodiversity shaped by diverse ecological conditions [8]. This biodiversity serves as a rich source of bioactive compounds, including proteins, carbohydrates, phenolic compounds, pigments, and fatty acids with distinctive chemical properties [7,9]. Among these, marine-derived bioactive peptides (BAPs) have demonstrated exceptional pharmaceutical and nutritional potential [10,11]. Their advantages include a low risk of adverse effects and cost-effective production [12]. Structurally, BAPs are protein fragments characterized by specific amide or peptide bonds and endowed with multiple bioactivities that support essential metabolic functions in the human body [12]. These attributes have fueled growing interest in the discovery and development of novel BAPs with therapeutic applications.

Ark shell (*Scapharca subcrenata*), a protein-rich and economically valuable seafood in Korea, Japan, and China, has recently gained attention for its diverse health benefits. Traditionally consumed for its nutritional value, recent research has highlighted its potential as a source of BAPs with therapeutic applications. Our team previously investigated ark shell protein hydrolysates produced through enzymatic hydrolysis and derived peptides, demonstrating their significant anti-inflammatory, osteogenic, anti-adipogenic, and anti-osteoporotic effects in both in vitro and in vivo models [13,14,15]. Among these bioactive compounds, the ark shell-derived peptides AWLNH (P3) and PHDL (P4) have been reported to exhibit anti-osteoporotic activities in several studies [13,14]. Furthermore, ark shell pepsin hydrolysates and their derived peptides, LLRLTDL or Bu1 and GYALPCDCL or Bu2, have been shown to effectively inhibit macrophage foam cell formation, a key process in atherosclerosis development [16,17]. Given the crucial role of foam cell formation in the progression of atherosclerosis, these findings suggest that ark shell-derived peptides could serve as promising candidates for cardiovascular protection. In this study, we aim to further investigate the therapeutic potential of AWLNH (P3) and PHDL (P4) by evaluating their inhibitory effects on oxLDL-induced macrophage foam cell formation by considering the multifunctional bioactivities of BAPs. Additionally, we assess their ability to modulate cholesterol metabolism and mitigate oxidative stress, two critical factors in the pathogenesis of atherosclerosis.

## 2. Results

### 2.1. P3 and P4 Peptides Inhibit oxLDL-Induced Lipid Accumulation in RAW264.7 Macrophages and hASMCs

The molecular structures of P3 and P4 peptides are illustrated in Figure 1. To assess potential cytotoxicity, an MTT assay was conducted in RAW264.7 macrophages prior to further experimentation. The results confirmed that P3 and P4 peptides exhibited no cytotoxic effects at the tested concentrations, which were subsequently selected for downstream analyses (Figure 2A). To evaluate the impact of P3 and P4 peptides on oxLDL-induced foam cell formation, intracellular lipid accumulation was assessed in both RAW264.7 macrophages and hASMCs. OxLDL treatment significantly increased (*p* < 0.05) intracellular lipid accumulation in RAW264.7 macrophages and hASMCs, respectively. However, P3 and P4 peptides effectively attenuated this lipid accumulation, reducing it by 68.3 ± 1.6% and 66.9 ± 2.2% in RAW264.7 macrophages and by 75.1 ± 1.9% and 80.4 ± 3.1% in hASMCs at a concentration of 200 µM (Figure 2B,C). These findings were further substantiated by Oil Red O staining, where representative images visually confirmed the inhibitory effects of P3 and P4 peptides on intracellular lipid accumulation (Figure 2D,E).

### 2.2. P3 and P4 Peptides Ameliorate Intracellular Cholesterol Levels in oxLDL-Treated RAW264.7 Macrophages

Since cholesterol metabolism is a key determinant of foam cell formation, we assessed the effects of P3 and P4 peptides on total cholesterol (TC), free cholesterol (FC), cholesterol ester (CE), and triglyceride (TG) levels. As illustrated in Figure 3A–D, both peptides significantly mitigated oxLDL-induced cholesterol accumulation. At a concentration of 200 µM, P3 peptide reduced TC, FC, and TG levels by 70.7 ± 1.0%, 71.4 ± 0.7%, and 66.7 ± 2.35%, respectively, while P4 peptide decreased FC and TG levels by 90.2 ± 1.5% and 71.7 ± 1.2%, respectively. Notably, P3 peptide completely abolished CE accumulation at 200 µM, whereas P4 peptide fully attenuated TC, FC, and CE levels at the same concentration, underscoring their potent modulatory effects on cholesterol homeostasis.

### 2.3. P3 and P4 Peptides’ Effect on Cholesterol Influx and Cholesterol Efflux and Related Transcription Factor Expressions in oxLDL-Treated RAW264.7 Macrophages

Elevated cholesterol influx and impaired cholesterol efflux are critical factors contributing to dysregulated cholesterol metabolism and increased foam cell formation. In this study, we assessed the effects of P3 and P4 peptides on these processes. At a concentration of 200 µM, P3 and P4 peptides significantly reduced cholesterol influx by 41.8 ± 2.9% and 46.6 ± 1.9%, respectively, compared to oxLDL treatment (Figure 4A). In contrast, both peptides enhanced cholesterol efflux, with P3 and P4 increasing efflux levels by 21.1 ± 1.5% and 20.3 ± 1.7%, respectively, at the same concentration (Figure 4B). Simvastatin and rosiglitazone were used as positive controls and showed strong effects in inhibiting cholesterol influx and increasing cholesterol efflux.

To further elucidate the molecular mechanisms, we examined the expression of genes associated with cholesterol influx and efflux. As shown in Figure 4C, both peptides markedly suppressed oxLDL-induced expression of the cholesterol influx-related genes CD36 and SR-A1 while upregulating the expression of the cholesterol efflux transporters ABCA1 and ABCG1. Given the central role of PPAR-γ in promoting cholesterol efflux, and considering that our peptides predominantly enhanced efflux over influx, we further investigated their impact on PPAR-γ and LXR-α expression. Western blot analysis (Figure 4D) demonstrated a dose-dependent upregulation of these transcription factors upon P3 and P4 treatment, reinforcing their role in facilitating cholesterol efflux and mitigating foam cell formation.

### 2.4. P3 and P4 Peptides Suppressed Oxidative Stress in oxLDL-Treated RAW264.7 Macrophages

Given the pivotal role of oxidative stress in foam cell formation, we evaluated the effects of P3 and P4 peptides on intracellular ROS production. As shown in Figure 5A, oxLDL exposure significantly increased ROS levels in RAW264.7 macrophages compared to untreated controls. However, treatment with P3 and P4 peptides effectively mitigated this oxidative stress, with P3 reducing ROS levels by 43.7 ± 2.4% and P4 by 55.2 ± 4.9%, both at a concentration of 200 µM. Representative images further corroborated these findings, visually confirming the inhibitory effects of P3 and P4 peptides on ROS generation (Figure 5B).

To delineate the molecular mechanisms underlying this antioxidative effect, we examined the activation of the Nrf2/HO-1 signaling pathway, a critical regulator of cellular defense against oxidative stress. Western blot analysis (Figure 5C) demonstrated a dose-dependent upregulation of HO-1 expression following P3 and P4 treatment. Since Nrf2 serves as a key transcriptional activator of HO-1, we further investigated its nuclear translocation. Consistent with our expectations, peptide treatment enhanced Nrf2 activation, as evidenced by increased nuclear Nrf2 accumulation and a concomitant decrease in cytoplasmic Nrf2 levels (Figure 5D).

### 2.5. Effects of HO-1 siRNA Transfection on oxLDL-Induced Foam Cells

Since HO-1 activation plays a crucial role in regulating oxidative stress, we hypothesized that the anti-foam cell formation effects of P3 and P4 peptides might be mediated through HO-1 activation. To validate this hypothesis, we transfected RAW264.7 macrophages with HO-1 siRNA and examined the impact of P3 and P4 peptides on foam cell formation.

First, we assessed HO-1 expression following siRNA transfection. As shown in Figure 6A, HO-1 siRNA significantly (*p* < 0.05) suppressed HO-1 expression in P3- and P4-treated macrophages, confirming successful knockdown. Next, we evaluated the effect of HO-1 silencing on cholesterol efflux-related transcription factors, including PPAR-γ and LXR-α. Interestingly, HO-1 knockdown reversed the upregulation of PPAR-γ and LXR-α induced by P3 and P4 peptides.

We then investigated the effect of HO-1 siRNA on cholesterol flux-associated proteins. As illustrated in Figure 6B, HO-1 silencing significantly (*p* < 0.05) inhibited the P3- and P4-mediated upregulation of cholesterol efflux transporters ABCA1 and ABCG1. Conversely, HO-1 knockdown restored the expression of cholesterol influx receptors CD36 and SR-A1, which were suppressed by P3 and P4 peptides.

Furthermore, we assessed the impact of HO-1 knockdown on ROS generation. As shown in Figure 7A, HO-1 siRNA abolished the ROS-reducing effects of P3 and P4 peptides, indicating that HO-1 activation is critical for their antioxidative properties. Finally, we examined the effect of HO-1 silencing on foam cell formation. Notably, HO-1 knockdown restored lipid accumulation, counteracting the suppressive effects of P3 and P4 peptides on foam cell formation (Figure 7B). These findings collectively suggest that P3 and P4 peptides inhibit foam cell formation via HO-1 activation, which modulates oxidative stress, cholesterol metabolism, and lipid accumulation in macrophages.

## 3. Discussion

CVDs have emerged as a significant global health burden, with the development of effective therapies that minimize adverse effects remaining a considerable challenge [1]. Foam cell formation is a pivotal hallmark of atherosclerosis, a major contributor to CVD progression [4]. Consequently, identifying therapeutic agents that can effectively mitigate foam cell formation is of critical importance. Despite the widespread use of lipid-lowering agents such as statins and PCSK9 inhibitors, the search for more effective, targeted treatments remains ongoing. Marine-derived BAPs exhibit a broad spectrum of multifunctional bioactivities, making them promising candidates for the treatment of atherosclerosis [9,11]. Previous studies have shown that ark shell-derived peptides possess potent biological activities, which contribute to the regulation of various diseases [13,14,15]. In our earlier research, we demonstrated that ark shell-derived peptides LLRLTDL (Bu1) and GYALPCDCL (Bu2) exert significant anti-atherosclerotic effects in RAW 264.7 macrophages [16]. Building upon these findings, we aim to explore the bioactivity of ark shell-derived peptides AWLNH (P3) and PHDL (P4). These peptides, P3 and P4, have also demonstrated a range of biological effects, including anti-inflammatory properties. Given the multifunctional bioactivities of BAPs, this study seeks to comprehensively assess the anti-atherosclerotic potential of ark shell-derived P3 and P4 peptides by investigating their impact on foam cell formation and their associated antioxidative stress mechanisms.

The chemical structures of Bu1 and Bu2 are presented in Figure 8. With all four peptides (P3, P4, Bu1, and Bu2) synthesized by our research team after determining their sequences [13,18], the synthetic peptides were used to explore their biological functions and mechanisms of action. In our previous study, we demonstrated that Bu1 and Bu2 peptides exhibited significant anti-atherosclerotic effects, primarily through modulation of the PPAR-γ/LXR-α signaling axis, which is crucial for lipid metabolism and inflammation regulation. Consistent with these findings, our current investigation revealed that P3 and P4 also increased the expression of PPAR-γ and LXR-α, suggesting similar mechanisms in their potential anti-atherosclerotic action. Furthermore, in addition to the PPAR-γ/LXR-α signaling pathway, our study highlighted that P3 and P4 peptides exerted their effects through inhibition of foam cell formation via HO-1/Nrf2 signaling (Table 1).

Atherosclerotic lesion initiation and progression are profoundly influenced by macrophages, which, within the atherosclerotic plaque, exhibit a foamy appearance due to the internalization of modified LDLs, including oxLDLs [6,19]. Consequently, the suppression of oxLDL-induced macrophage foam cell formation is a critical therapeutic strategy for both the treatment and prevention of atherosclerosis. In this study, our findings demonstrate that P3 and P4 peptides significantly reduce oxLDL internalization and decrease foam cell formation, highlighting their inhibitory effects on oxLDL-induced foam cell formation in RAW264.7 macrophages. To enhance the clinical relevance of our results, we extended this investigation to human aortic smooth muscle cells (hASMCs). Consistent with our earlier observations, P3 and P4 peptides also effectively suppressed intracellular lipid accumulation in hASMCs, further supporting their potential as therapeutic agents in atherosclerosis.

The formation of foam cells is tightly regulated by a network of genes involved in cholesterol metabolism, and dysregulation of these pathways is a critical driver of atherosclerosis development [4,20,21]. Specifically, genes associated with cholesterol influx, such as CD36 and SR-A1, facilitate the internalization of oxLDLs into macrophages [22,23,24,25,26]. Once internalized, oxLDLs are hydrolyzed by lysosomal acid lipase, resulting in the release of free fatty acids and free cholesterol. These free molecules are subsequently re-esterified into cholesterol esters, the primary storage lipids within the endoplasmic reticulum, by acyl-coenzyme A:cholesterol acetyltransferase (ACAT) [23]. Under normal physiological conditions, macrophages maintain lipid homeostasis by efficiently removing excess lipids via the cholesterol efflux pathway, a process primarily mediated by ATP-binding cassette (ABC) transporters such as ABCA-1 and ABCG-1 [27,28]. However, oxLDLs exacerbate lipid accumulation by upregulating cholesterol influx while concurrently impairing the function of cholesterol efflux mechanisms. Our results indicate that treatment with P3 and P4 peptides effectively mitigates oxLDL-induced cholesterol influx and promotes enhanced cholesterol efflux. Moreover, P3 and P4 peptides significantly attenuated oxLDL-induced accumulation of TC, FC, and CE in macrophages. In addition to their effects on cholesterol metabolism, these peptides also decreased TG levels, which are sequestered in lipid droplets, further highlighting their potential as modulators of lipid metabolism in the context of atherosclerosis.

Cholesterol efflux has been shown to be regulated by the activation of PPAR-γ [16]. In light of this, we investigated the effects of the transcription factors PPAR-γ and LXR-α on cholesterol efflux. Both LXR-α and PPAR-γ play key roles in regulating cholesterol metabolism by upregulating the expression of ABCA-1 and ABCG-1, which facilitate the removal of excess cholesterol from cells [29,30,31]. Previous studies, including our own, have highlighted the importance of these pathways in maintaining lipid homeostasis, prompting us to explore the effects of P3 and P4 peptides on PPAR-γ and LXR-α activation [16]. PPAR-γ, while involved in cholesterol metabolism, has a dual role, also enhancing the expression of genes linked to cholesterol influx, such as CD36. However, its net effect tends to favor cholesterol efflux [32]. Supporting this, our data show that P3 and P4 peptides activate both PPAR-γ and LXR-α signaling pathways. Taken together, these findings suggest that P3 and P4 peptides regulate cholesterol metabolism during oxLDL-induced foam cell formation by activating the PPAR-γ/LXR-α signaling axis, thereby enhancing cholesterol efflux and potentially mitigating the development of atherosclerosis.

Atherosclerosis is intimately associated with oxidative stress, which plays a crucial role in its initiation and progression [33,34]. In this context, the present study aimed to evaluate the impact of oxidative stress in oxLDL-induced RAW264.7 macrophages following treatment with P3 and P4 peptides. The HO-1/Nrf2 signaling pathway, a well-established regulator of antioxidant responses, was chosen for investigation due to its central role in cellular protection against oxidative damage. HO-1, a key detoxifying enzyme, is induced under conditions of oxidative stress and catalyzes the degradation of heme into ferrous, carbon monoxide, and biliverdin end products with potent antioxidant properties [35]. The activation of HO-1 is tightly regulated by Nrf2, a transcription factor that, under basal conditions, is sequestered in the cytoplasm. In response to oxidative stress, Nrf2 translocates to the nucleus, where it promotes the expression of several antioxidant genes, including HO-1 [36,37]. Our findings demonstrate that P3 and P4 peptides significantly upregulate HO-1 expression and promote the nuclear translocation of Nrf2, suggesting their potential to mitigate oxidative stress. This activation of the HO-1/Nrf2 axis underscores the peptides’ antioxidant properties, indicating their therapeutic potential in reducing oxidative stress-related damage in atherosclerosis.

Moreover, P3 and P4 peptides effectively suppressed the generation of ROS, further reinforcing their role in mitigating oxidative stress [38]. To substantiate these findings, we utilized HO-1 siRNA transfection, which allowed us to assess the specific involvement of HO-1 in the observed effects. HO-1 knockdown reversed the peptide-induced upregulation of HO-1 expression and the reduction in ROS generation, confirming the critical role of HO-1 in mediating the antioxidant effects of P3 and P4 peptides.

In addition, we explored the impact of HO-1 on cholesterol metabolism by analyzing the expression of genes involved in cholesterol influx and efflux following HO-1 siRNA treatment. As expected, HO-1 silencing reversed the peptides’ effects, leading to an increase in CD36 and SR-A1 expression, which is associated with cholesterol influx, while simultaneously reducing the expression of ABCA1 and ABCG1, key genes involved in cholesterol efflux. Moreover, HO-1 siRNA treatment also reversed the peptides’ effects on the expression of critical transcription factors, including PPAR-γ and LXR-α. Finally, we assessed the functional consequences of HO-1 silencing by visualizing foam cell formation through lipid accumulation analysis. HO-1 siRNA treatment reversed the peptide-mediated reduction in lipid accumulation, further supporting the pivotal role of HO-1 in regulating cholesterol metabolism and foam cell formation. Taken together, these results underscore that P3 and P4 peptides exert their beneficial effects on oxidative stress, cholesterol metabolism, and foam cell formation through the modulation of the HO-1/Nrf2 signaling pathway.

## 4. Materials and Methods

### 4.1. Materials

Chemically synthesized AWLNH (P3) and PHDL (P4) peptides were obtained from Peptron Inc. (Daejeon, Republic of Korea) based on the sequences identified in our previous study [13]. Human plasma LDLs were purchased from LEE BioSolutions (360-10, Lee BioSolution, Maryland Heights, MO, USA). Cell culture reagents were purchased from Gibco BRL (Grand Island, NY, USA). Primary and secondary antibodies were obtained from Santa Cruz Biotechnology Inc. (Santa Cruz, CA, USA), NovusBio^®^, USA, and Abcam (Dawinbio Inc., Cambrigde, UK). All other reagents utilized in this study were obtained from Sigma-Aldrich (St. Louis, MO, USA).

### 4.2. Oxidation of LDLs and Determination of Thiobarbituric Acid-Reactive Substances (TBARSs)

LDLs (1 mg/mL) were oxidized with CuSO_4_ (10 μM) for 4 h at 37 °C and the oxidation level was determined by performing a TBARS assay, as in our previous study [39].

### 4.3. Cell Culture and Treatment

RAW264.7 macrophages (American Type Culture Collection, Manassas, VA, USA) were cultured in DMEM culture medium under standard conditions. For the experiments, cell densities of 1 × 10^5^ cells/mL were seeded in 96-well plates, and 1 × 10^6^ cells/mL were seeded in 6 cm culture dishes. The macrophages were treated with P3 or P4 peptides (10–200 µM) for 1 h, followed by a 24 h exposure to oxLDL. hASMCs, obtained from ScienCell Research Laboratories (Carlsbad, CA, USA), were cultured in smooth muscle cell medium (SMCM, 1101, ScienCell Research Laboratories) under standard conditions. Both macrophages and hASMCs were treated with P3 or P4 peptides for 1 h, followed by a 24 h treatment with oxLDL.

### 4.4. MTT Assay

Cytotoxicity of P3 and P4 (10~200 µM) in RAW264.7 macrophages was evaluated using the MTT assay, as described previously [36].

### 4.5. ORO Staining Assay

RAW264.7 macrophages were treated with P3 or P4 peptides (10~200 µM) for 1 h followed by 24 h treatment with oxLDL, and an ORO staining assay was performed, as previously reported [36].

### 4.6. Determination of TC, FC, CE, and TG Content

RAW264.7 macrophages were seeded in 12-well plates and treated with P3 and P4 samples (10~200 µM), as described above. TC and FC were determined using BioVision (BioVision Inc., Mountain View, CA, USA) total cholesterol and cholesteryl ester colorimetric assay kit II, according to the manufacturer’s instructions. The protein concentration of treated cells was determined by BCA assay. Cholesterol ester amount was calculated by subtracting FC from TC. Cellular TG level was measured by a commercially available colorimetric TG assay kit (Biomax, Seoul, Republic of Korea).

### 4.7. Determination of Cholesterol Influx and Efflux

RAW264.7 cells were treated with P3 or P4 peptides (10~200 µM) for 1 h followed by 24 h treatment with oxLDLs in 96-black-well plates and cholesterol influx and efflux effects were determined using 25-NBD-cholesterol, as reported in previous studies [40].

### 4.8. Western Blot Analysis

Western blot analysis was performed according to standard protocol after the macrophages were treated with P3 or P4 peptides (10~200 µM) for 1 h followed by oxLDL treatment for another 24 h and whole-cell lysates were prepared using RIPA buffer in the presence of protease and phosphatase inhibitors (Roche Diagnostics, Seoul, Republic of Korea). A chemiluminescence ECL assay kit (Life Technologies, Seoul, Republic of Korea) was used to visualize the bands which were imaged on a Davinch-Chemi Imager™ (CAS400SM, Core Bio, Seoul, Republic of Korea).

### 4.9. siRNA Transfection

RAW264.7 macrophages (3 × 10^5^) were seeded in 6 cm well plates without antibiotics one day prior to the transfection. Macrophages were transiently transfected with 100 nM of HO-1 siRNA or negative control by using Lipofectamine^®^ 2000 (Invitrogen; Thermo Fisher Scientific, Inc., Waltham, MA, USA) for 24 h according to the manufacturer’s instructions. After that, macrophages were treated with 200 µM of P3 or P4 peptides for 1 h followed by oxLDL treatment for another 24 h.

### 4.10. Statistical Analysis

Sigma Plot 12.0 (Systat Software Inc., San Jose, CA, USA) was used to carry out a one-way ANOVA and results were presented as means ± S.D. (*n* = 3). A Student *t*-test was performed and values with *p* < 0.05 were regarded as statistically significant.

## 5. Conclusions

Marine-derived bioactive peptides (BAPs) are rich in therapeutic properties and have been widely explored for their potential in treating various diseases. In this study, we demonstrated that ark shell-derived peptides P3 and P4 exert significant anti-atherosclerotic effects by inhibiting foam cell formation. Specifically, P3 and P4 peptides effectively suppressed intracellular lipid accumulation in oxLDL-induced RAW264.7 macrophages and hASMCs. Mechanistically, these peptides reduced cholesterol influx while enhancing cholesterol efflux through the activation of the PPAR-γ/LXR-α signaling pathway. Additionally, P3 and P4 exhibited potent antioxidative properties by activating the HO-1/Nrf2 pathway, leading to a significant reduction in ROS generation. The critical role of HO-1 in foam cell formation inhibition was further confirmed through HO-1 siRNA transfection, which reversed the peptides’ protective effects on cholesterol metabolism, transcriptional regulation, and foam cell formation. These findings underscore the intricate relationship between oxidative stress and lipid homeostasis in the progression of atherosclerosis, suggesting that targeting the HO-1/Nrf2 axis could be a promising therapeutic strategy. Overall, our study highlights P3 and P4 as potential candidates for the development of novel anti-atherosclerotic therapies.

## Figures and Tables

**Figure 1 marinedrugs-23-00111-f001:**
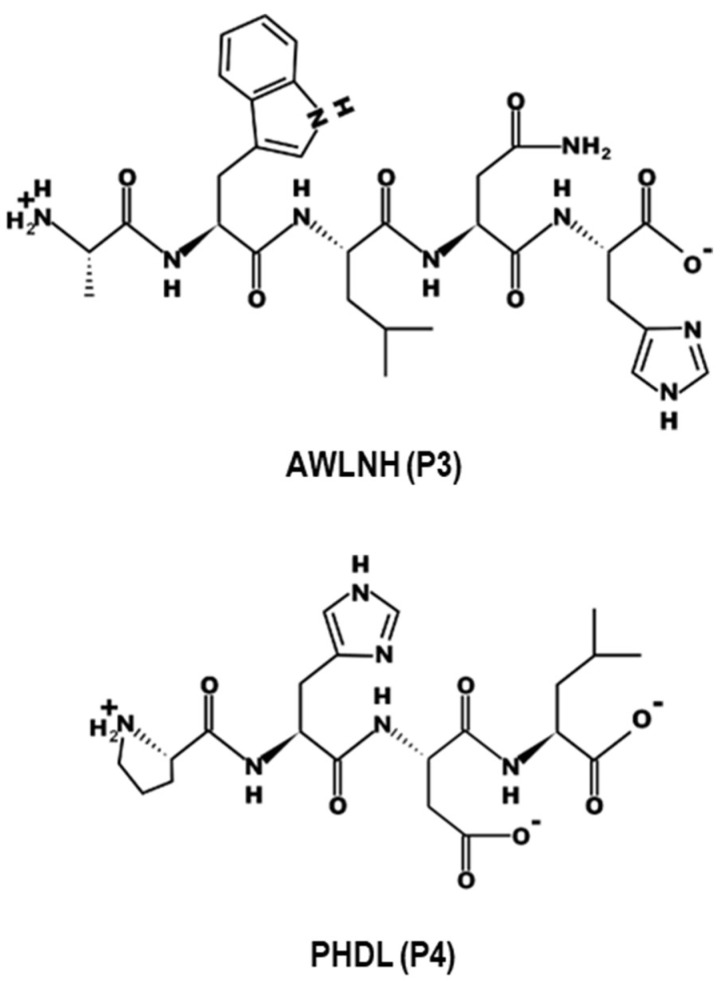
Chemical structures of AWLNH (P3) and PHDL (P4).

**Figure 2 marinedrugs-23-00111-f002:**
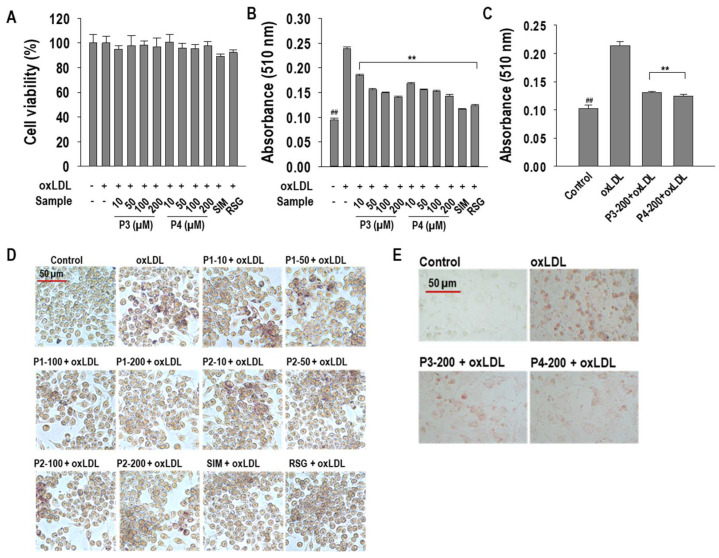
(**A**) Cell viability; quantitative analysis of intracellular lipid accumulation inhibition in (**B**) RAW264.7 macrophages and (**C**) hASMCs; and qualitative evaluation of AWLNH (P3) and PHDL (P4) in (**D**) oxLDL-treated RAW264.7 macrophages (40× magnification) and (**E**) hASMCs (20× magnification). For the MTT assay, macrophages were exposed to P3, P4, or positive controls (10 µM), including simvastatin (SIM) or rosiglitazone (RSG), along with oxLDLs (50 µg/mL). In Oil Red O (ORO) staining experiments, cells were pretreated with P3, P4, or positive controls (10 µM), including SIM or RSG, for 1 h before oxLDL exposure for 24 h. Data are presented as mean ± S.D. from three independent experiments (*n* = 3). Statistical significance is indicated as *** p* < 0.001 versus the oxLDL-treated group and *^##^ p* < 0.001 versus the non-treated group. Numbers in the images denote concentrations in µM.

**Figure 3 marinedrugs-23-00111-f003:**
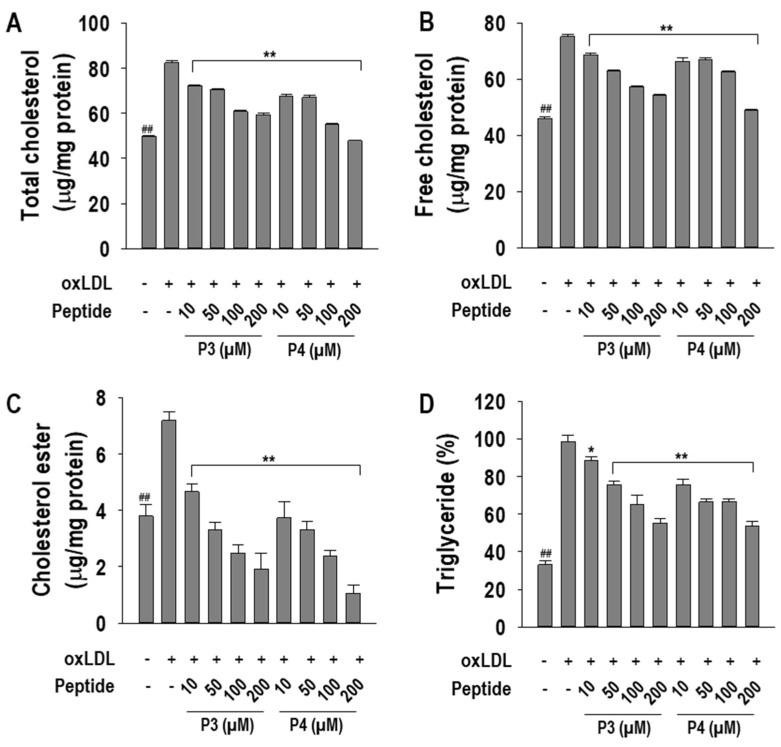
The effects of 10–200 µM concentrations of AWLNH (P3) and PHDL (P4) peptides on (**A**) total cholesterol, (**B**) free cholesterol, (**C**) cholesterol ester, and (**D**) triglyceride content in oxLDL-treated RAW264.7 macrophages. Macrophages were treated with P3 and P4 peptides for 1 h, followed by a 24 h treatment with oxLDL. The results are based on three independent experiments (*n* = 3), with data expressed as mean ± S.D. Significant differences are indicated by ** p* < 0.05, *** p* < 0.001, comparing the P3 and P4 peptide treatments to the oxLDL-treated group, and ^##^
*p* < 0.001, comparing the peptide treatments to the non-treated control group.

**Figure 4 marinedrugs-23-00111-f004:**
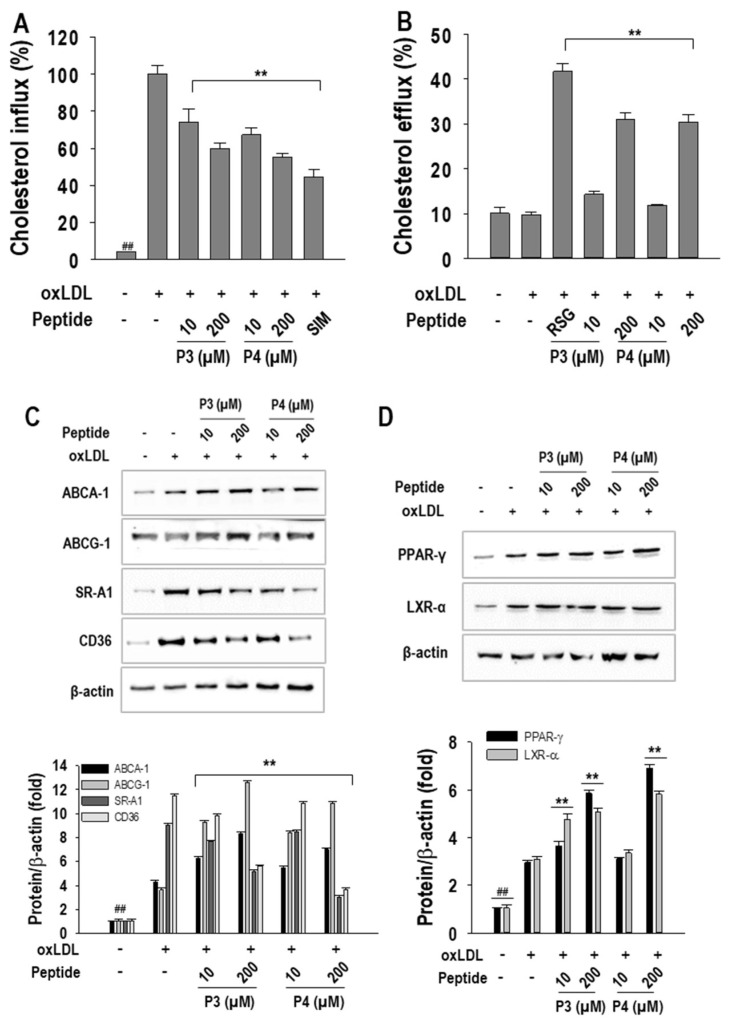
The effect of 10–200 µM concentrations of AWLNH (P3) and PHDL (P4) peptides on (**A**) cholesterol influx, (**B**) cholesterol efflux, (**C**) protein expressions of ABCA-1, ABCG-1, SR-A1, and CD36 and (**D**) PPAR-γ and LXR-α expression in oxLDL-treated RAW264.7 macrophages. Macrophages were pretreated with P3 and P4 peptides for 1 h, followed by a 24 h treatment with oxLDL. The results are based on three independent experiments (*n* = 3), with data presented as mean ± S.D. Statistical significance is denoted as *** p* < 0.001 when comparing the peptide treatments to the oxLDL-treated group and ^##^
*p* < 0.001 when comparing to the non-treated group.

**Figure 5 marinedrugs-23-00111-f005:**
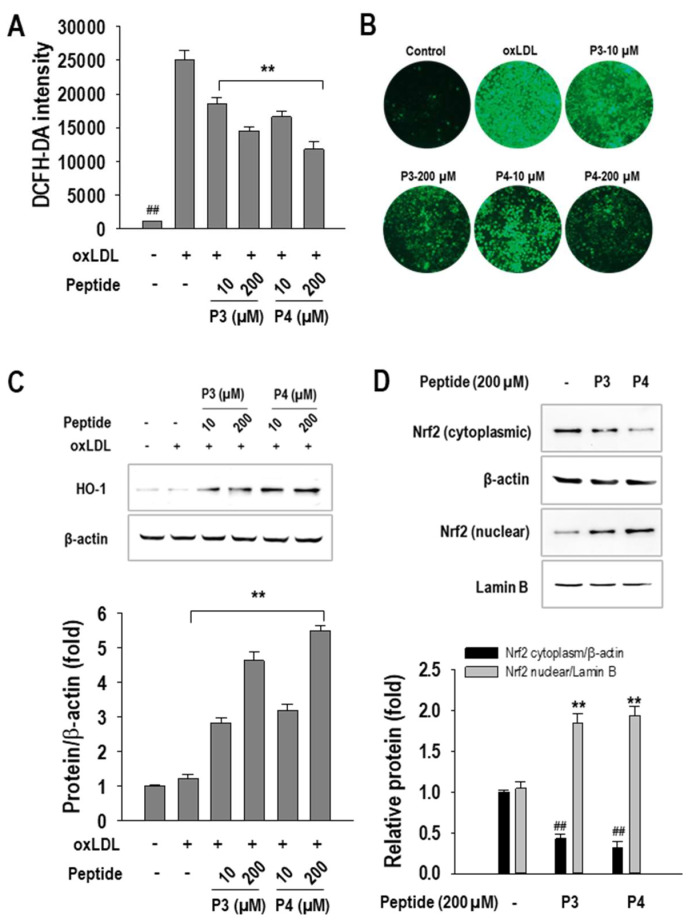
The effect of 10–200 µM concentrations of AWLNH (P3) and PHDL (P4) peptides on (**A**) quantitative ROS generation, (**B**) qualitative ROS generation (20× magnification), and (**C**) HO-1 expression in oxLDL-treated RAW264.7 macrophages, as well as (**D**) Nrf2 activation. For ROS and HO-1 analysis, macrophages were treated with P3 and P4 peptides for 1 h, followed by oxLDL exposure for 24 h. For Nrf2 activation, macrophages were treated with P3 and P4 peptides for 1 h. Data are presented as the mean ± S.D. from three independent experiments (*n* = 3). Statistical significance is indicated by *** p* < 0.001 compared to the oxLDL-treated group, ^##^
*p* < 0.001 compared to the non-treated group in ROS and HO-1 analysis, *** p* < 0.001 compared to the nuclear fraction of the non-treated group, and ^##^
*p* < 0.001 compared to the cytoplasmic fraction of the non-treated group. The numbers in the images represent peptide concentrations in µM.

**Figure 6 marinedrugs-23-00111-f006:**
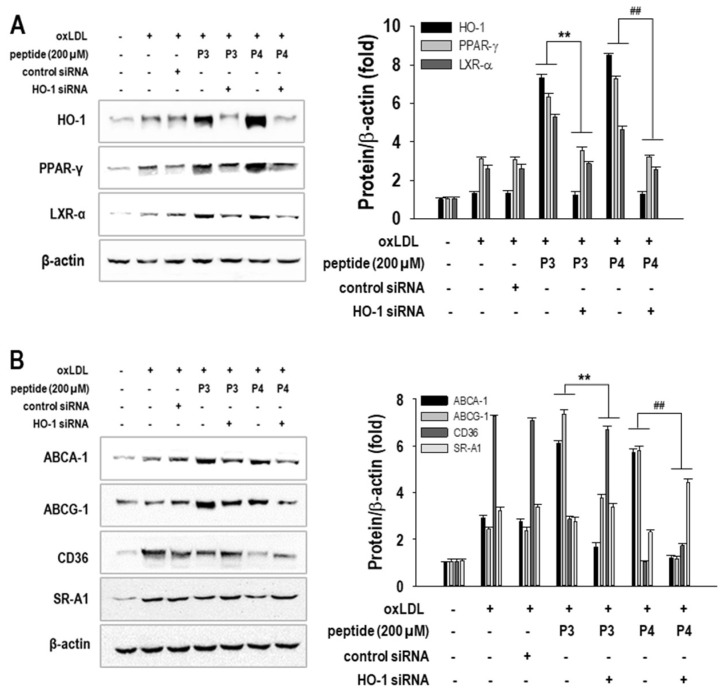
The effect of 200 µM concentrations of AWLNH (P3) and PHDL (P4) peptides on (**A**) HO-1, PPAR-γ, and LXR-α expressions, and (**B**) ABCA-1, ABCG-1, CD36, and SR-A1 expressions with HO-1 siRNA transfection was assessed. RAW264.7 macrophages were transiently transfected with HO-1 siRNA or a negative control using Lipofectamine^®^ 2000 for 24 h. The cells were then treated with 200 µM of P3 or P4 peptides for 1 h, followed by a 24 h treatment with oxLDLs (50 µg/mL). Data represent the mean ± S.D. from three independent experiments (*n* = 3). Statistical significance is indicated by ** *p* < 0.001 compared to the P3-treated group and ^##^
*p* < 0.001 compared to the P4-treated group in siRNA transfection.

**Figure 7 marinedrugs-23-00111-f007:**
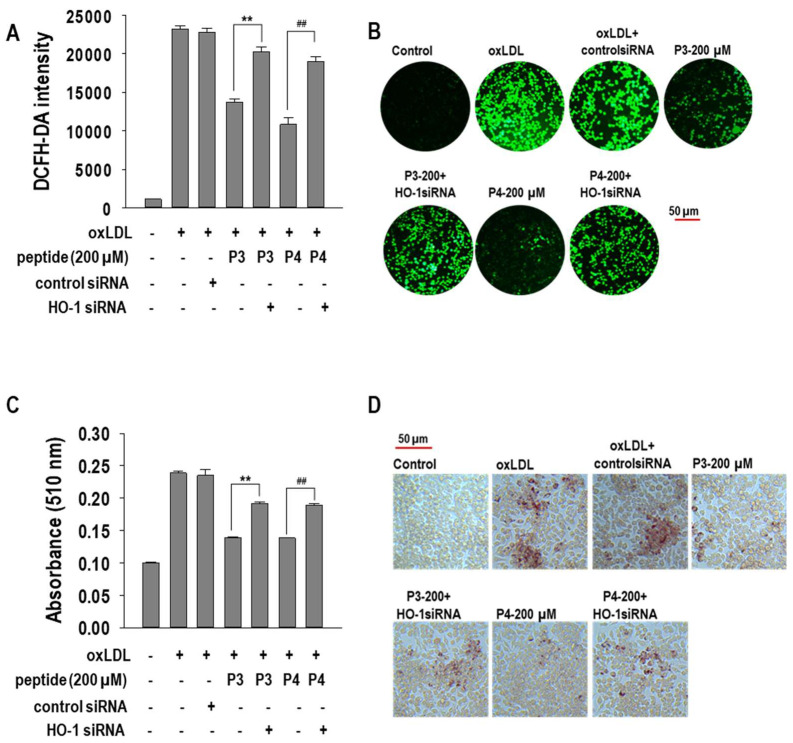
The effect of 200 µM concentrations of AWLNH (P3) and PHDL (P4) peptides on ROS generation, (**A**) quantitatively and (**B**) qualitatively (40× magnification), and intracellular lipid accumulation, (**C**) quantitatively and (**D**) qualitatively (20× magnification), with HO-1 siRNA transfection was assessed. RAW264.7 macrophages were transiently transfected with HO-1 siRNA or a negative control using Lipofectamine^®^ 2000 for 24 h. The cells were then treated with 200 µM of P3 or P4 peptides for 1 h, followed by a 24 h treatment with oxLDLs (50 µg/mL). Data represent the mean ± S.D. from three independent experiments (*n* = 3). Statistical significance is indicated by ** *p* < 0.001 compared to the P3-treated group and ^##^
*p* < 0.001 compared to the P4-treated group in siRNA transfection.

**Figure 8 marinedrugs-23-00111-f008:**
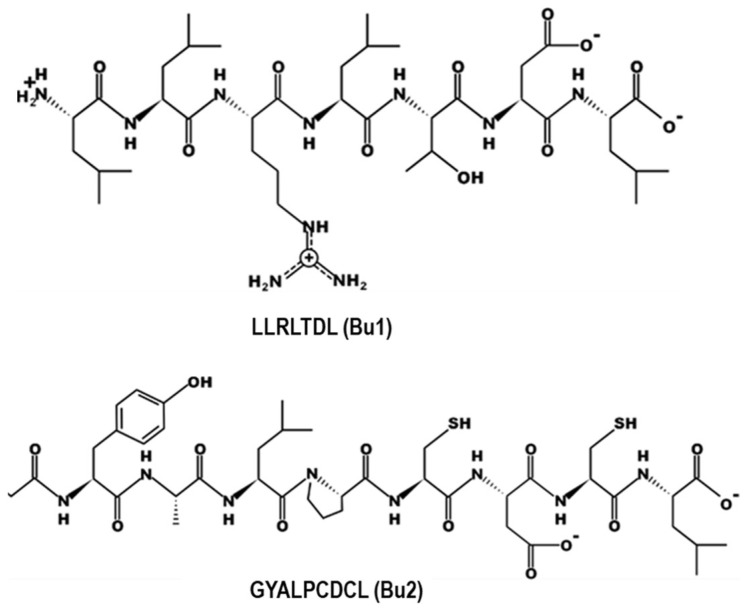
Chemical structures of LLRLTDL (Bu1) and GYALPCDCL (Bu2).

**Table 1 marinedrugs-23-00111-t001:** Health benefits of BAPs derived from ark shells.

Peptide Sequences	Originally Observed Bioactivity	Foam Cell Formation Inhibitory Action
AWLNH (P3) PHDL (P4)	Osteogenesis and anti-osteoporotic activity [13]	HO-1/Nrf2 signaling
LLRLTDL (Bu1) GYALPCDCL (Bu2)	Anti-adipogenesis [18]	PPAR-γ/LXR-α signaling

## Data Availability

Data are contained within the article.

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
