# Peer review of "Ark Shell-Derived Peptides AWLNH (P3) and PHDL (P4) Mitigate Foam Cell Formation by Modulating Cholesterol Metabolism and HO-1/Nrf2-Mediated Oxidative Stress in Atherosclerosis"

_marinedrugs, 2025, doi:10.3390/md23030111_

Round 1
Reviewer 1 Report
Comments and Suggestions for Authors
This study elucidates the anti-atherosclerotic potential of AWLNH (P3) and PHDL (P4) peptides by assessing their effects on foam cell formation, lipid metabolism, and oxidative stress regulation. P3 and P4 effectively suppressed intracellular lipid accumulation in RAW264.7 macrophages and human aortic smooth muscle cells (hASMCs), thereby mitigating foam cell formation.These findings highlight P3 and P4 peptides as promising therapeutic agents for atherosclerosis by concurrently targeting foam cell formation, cholesterol dysregulation, and oxidative stress, warranting further exploration for potential clinical applications.
This manuscript can be accepted following minor revisions.
1.What is the basis for selecting the drug dosage? Was it supported by preliminary experiments or existing literature?
2.The current reference formatting does not adhere to standard guidelines. Please revise the references according to established academic standards.
3.The author should redraw the chemical structures of AWLNH (P3) and PHDL (P4) use ACS 1996.
Comments on the Quality of English Language
The English can be polished in the revised manuscript.
Author Response
This study elucidates the anti-atherosclerotic potential of AWLNH (P3) and PHDL (P4) peptides by assessing their effects on foam cell formation, lipid metabolism, and oxidative stress regulation. P3 and P4 effectively suppressed intracellular lipid accumulation in RAW264.7 macrophages and human aortic smooth muscle cells (hASMCs), thereby mitigating foam cell formation. These findings highlight P3 and P4 peptides as promising therapeutic agents for atherosclerosis by concurrently targeting foam cell formation, cholesterol dysregulation, and oxidative stress, warranting further exploration for potential clinical applications.
Response: Thank you very much for your valuable comments.
(1) What is the basis for selecting the drug dosage? Was it supported by preliminary experiments or existing literature?
Response: Thank you for your comment. We selected concentrations of 10~200 µM for AWLNH (P3) and PHDL (P4) based on cell viability studies and preliminary Oil Red O experiments. These concentrations were chosen as they did not exhibit cytotoxicity and demonstrated a promising effect in reducing intracellular lipid accumulation.
(2) The current reference formatting does not adhere to standard guidelines. Please revise the references according to established academic standards.
Response: Thank you for your comment. We have formatted according to the standard.
(3) The author should redraw the chemical structures of AWLNH (P3) and PHDL (P4) use ACS 1996.
Response: Thank you for your suggestion. We have redrawn the chemical structures.
Reviewer 2 Report
Comments and Suggestions for Authors
- Line 101 Figure 1 needs to be written as "Figure 1A"
- Since this is continuation work, include chemical structures of other Ark Shell BAPs (e.g., Bu1 and Bu2, etc.) in Figure 1A. Also provide literature sources for the peptides. Who isolated or extracted these molecules?
- Since this is continuation work, add a table of P3, P4 versus Bu1 and Bu2 peptides in the results section. This table should contrast differences in potencies, structure, etc. essentially are P3 and P4 more potent than Bu1 and Bu2?
- Please note that Ref. #16 was improperly cited. This citation must be corrected.
Author Response
Reviewer #2
Response: Thank you very much for your valuable comments.
MAJOR COMMENTS
- Line 101 Figure 1 needs to be written as "Figure 1A"
Response: Thank you for your comment. We changed Fig. 1A in line 86 to Fig. 1.
- Since this is continuation work, include chemical structures of other Ark Shell BAPs (e.g., Bu1 and Bu2, etc.) in Figure 1A. Also provide literature sources for the peptides. Who isolated or extracted these molecules?
Response: Thank you for your inquiry. As stated, the peptides AWLNH (P3) and PHDL (P4) were identified in our previous study (https://doi.org/10.1016/j.taap.2019.114779) and we mentioned it in line 357-358. For the present study, the sequences of these peptides were provided to Peptron Inc. for chemical synthesis. Bu1 and Bu2 structures were added in Figure 8.
- Since this is continuation work, add a table of P3, P4 versus Bu1 and Bu2 peptides in the results section. This table should contrast differences in potencies, structure, etc. essentially are P3 and P4 more potent than Bu1 and Bu2?
Response: Thank you for your comment and suggestion. As mentioned in the discussion section (lines 334-352), all four peptides (P3, P4, Bu1, and Bu2) from ark shell hydrolysates showed promising anti-atherosclerotic effects, each through its own mechanism. These effects were primarily mediated by the modulation of the PPAR-γ/LXR-α signaling pathway, which plays a key role in lipid metabolism and inflammation regulation. Consistent with these findings, our current investigation revealed that P3 and P4 also increased the expression of PPAR-γ and LXR-α, suggesting similar mechanisms in their potential anti-atherosclerotic action. Additionally, P3 and P4 peptides were found to inhibit foam cell formation through the HO-1/Nrf2 signaling pathway. Since all four peptides demonstrated promising effects through different mechanisms, it is difficult to establish a direct comparison of their potency at this stage. As a result, we did not include a table comparing their relative effects. However, we have included their chemical structures in Figure 8 as requested. We hope this explanation addresses your comment. Please let us know if any further modifications are needed8.
- Please note that Ref. #16 was improperly cited. This citation must be corrected.
Response: Thank you for your comment. We corrected the citation.
Round 2
Reviewer 2 Report
Comments and Suggestions for Authors
Authors, please attend to the following items:
- lines 335 - 340 are not needed (please delete them) because they simply repeat what is already stated in in sentence 244-248.
- Cut lines 340 "The chemical structs......" through 351 plus figure 8 and paste all of that as a new paragraph between lines 252 and 253.
- Since this is a continuation project, please provide a table (in the results section) contrasting the activities of P3, P4, vs Bu1 & Bu2. Please do not ignore this important point. It helps distinguish this work from the previous work, otherwise this is simply a screening project.
Comments on the Quality of English Language
None.
Author Response
Comments 1
lines 335 - 340 are not needed (please delete them) because they simply repeat what is already stated in in sentence 244-248.
Cut lines 340 "The chemical structs......" through 351 plus figure 8 and paste all of that as a new paragraph between lines 252 and 253.
Response: Thank you for your suggestion. We did it.
Comments 2
Since this is a continuation project, please provide a table (in the results section) contrasting the activities of P3, P4, vs Bu1 & Bu2. Please do not ignore this important point. It helps distinguish this work from the previous work, otherwise this is simply a screening project.
Response: Yes, we made Table 1. Health-benefits of BAPs derived from ark shell. This is provided in line 234 in Result section.
Round 3
Reviewer 2 Report
Comments and Suggestions for Authors
Last corrections on my part:
- Line 73 needs to be corrected to LLRLTDL or Bu1 and GYALPCDCL or Bu2,
- Table 1 does not fully address my suggestion but will suffice. Also, in table 1, note that P3 is repeated for both peptides.
Comments on the Quality of English Language
None....
Author Response
Last corrections on my part:
Response: Thank you for your time.
Q1: Line 73 needs to be corrected to LLRLTDL or Bu1 and GYALPCDCL or Bu2,
Response: Yes. We revised it accordingly.
Q2: Table 1 does not fully address my suggestion but will suffice. Also, in table 1, note that P3 is repeated for both peptides.
Response: Thank you for your valuable comment. We revised P3 to P4.